# Investigation of The Effects of Oxytocin Administration Timing on Postpartum Hemorrhage during Cesarean Section

**DOI:** 10.3390/medicina59020222

**Published:** 2023-01-24

**Authors:** Soner Gök, Mehmet Babür Kaleli

**Affiliations:** 1Department of Obstetrics and Gynecology, School of Medicine, Pamukkale University, Denizli 20160, Turkey; 2Department of Perinatology, School of Medicine, Pamukkale University, Denizli 20160, Turkey

**Keywords:** cesarean section, hemoglobin, hematocrit, oxytocin, postpartum hemorrhage

## Abstract

*Background and Objectives*: To determine and compare the effects of the timing of oxytocin administration (routinely used for intraoperative uterotonic purposes in cesarean section (CS) deliveries in our clinic) on the severity of postpartum hemorrhage following CS. *Materials and Methods*: All study participants (*n* = 216) had previous cesarean deliveries, were 38–40 weeks pregnant, and had CS planned under elective conditions. The cases were randomly divided into two groups: one group (*n* = 108) receiving oxytocin administration before the removal of the placenta (AOBRP) and another group (*n* = 108) receiving oxytocin administration after the removal of the placenta (AOARP). In all cases, the placenta was removed using the manual traction method. The standard dose of oxytocin is administered as an intravenous (IV) push of 3 international units (IU); simultaneously, 10 IU of oxytocin is added to 1000 cc isotonic fluid and given as an IV infusion at a rate of 250 cc/h. All methods and procedures applied to both groups were identical, except for the timing of administration of the standard oxytocin dose. Age, body mass index (BMI), parity, gestational week, preoperative hemoglobin (HB) and hematocrit (HTC), postoperative 6th and 24th hour HB-HTC, intraoperative hemorrhage, additional uterotonic need during cesarean section, postoperative hemorrhage (number of pads), need for blood transfusion during or after cesarean section, cesarean section time, and postpartum newborn baby weight were evaluated. *Results*: Age (year), BMI (kg/m^2^), parity, gestational week, surgical time, and newborn weight (g) did not differ between the groups (*p* > 0.05). The AOBRP group had significantly higher postoperative 6th hour HB and HTC and postoperative 24th hour HB and HTC values (*p* < 0.05). The intraoperative hemorrhage level was higher in the AOARP group (*p* = 0.000). *Conclusions*: The administration of oxytocin before placenta removal did not change the volume of bleeding in the postoperative period but significantly reduced the volume of bleeding in the intraoperative period. Therefore, in the postoperative period, the HB and HTC values of the AOBRP group were higher than those of the AOARP group.

## 1. Introduction

Childbirth can be divided into vaginal births and cesarean (CS) births. Recent studies have shown that cesarean delivery rates have increased worldwide [1]. According to data published by Turkey’s Ministry of Health in 2020, the percentage of the total number of cesarean sections to live births was 54.4% in 2019, and the percentage of the number of primary cesarean sections to live births was 26.5% [2].

Postpartum hemorrhage (PPH) during childbirth is a feared medical emergency [3]. The American College of Obstetricians and Gynecologists defines PPH as cumulative loss of at least 1000 mL of blood (or any volume if there are signs or symptoms of hypovolemia) within 24 h after birth [4]. When it is difficult to calculate the volume of the hemorrhage, it can also be defined as a decrease of more than 10% between pre- and postpartum hematocrit (HTC) levels [5,6]. The estimated incidence of PPH in women who delivered with CS is 3–15%, and 2–4% in women who delivered vaginally [7,8]. Worldwide, approximately 140,000 women die yearly due to PPH [9,10,11]. Because PPH is a life-threatening condition, obstetricians should be aware of the risk factors, prevention strategies, and actions required. Risk factors include fibroids, macrosomia, maternal obesity, polyhydramnios, inherited coagulation disorders (von Willebrand disease, hemophilia A, hemophilia B, and hemophilia C), multiple pregnancies, repeated cesarean sections, prolonged labor, prolonged third stage, grand multiparity, chorioamnionitis, PPH history, antepartum hemorrhage, and operative delivery for PPH [12,13,14].

Uterine atony is responsible for 50–80% of all PPH cases [15,16]. According to the World Health Organization (WHO), intravenous (IV) or intramuscular oxytocin is recommended for the prevention of PPH in all deliveries in settings in which multiple uterotonic options are available [16]. However, there is no clear recommendation as to the best time to administer oxytocin to prevent PPH in women who deliver by cesarean section. Current guidelines have various recommendations on doses, routes, and regimens for the administration of prophylactic oxytocin in CS, but most do not provide specific guidance on the timing of administration [6,17,18,19]. Some obstetricians administer prophylactic oxytocin at various times before fetal birth in CS, others administer oxytocin immediately after the baby is born and the umbilical cord is clamped, and another group delays oxytocin administration until the placenta separates from the uterus [20,21].

PPH, one of the most common causes of maternal morbidity and mortality, increases in parallel with an increase in cesarean section rates [22,23]. In this study, we aimed to determine the effects of the timing of standard-dose oxytocin administration (which we routinely apply during cesarean sections in our clinic) on the amount of intrapartum and postpartum bleeding. Therefore, we administered oxytocin at two different times: before and after placenta removal. In this way, it was planned to investigate the effects of oxytocin administration in equal doses on intraoperative and postoperative bleeding by changing only the time of administration. We hope that our results will contribute to the few studies conducted in this area. In addition, we believe that our study will have positive effects on maternal and newborn health.

## 2. Materials and Methods

### 2.1. Study Design and Patients

This research was carried out at Pamukkale University Hospital Gynecology and Obstetrics Clinic (Denizli, Turkey) between 15 December 2021 and 15 October 2022. The research protocol was approved by the Faculty of Medicine of Pamukkale University Ethics Committee (14 December 2021; 22).

All the cases participating in the study had previous cesarean deliveries, were 38–40 weeks pregnant, and planned for cesarean section under elective conditions. Pregnant women with systemic diseases, such as hypertension, diabetes, thyroid dysfunction, and major depression, as well as women with conditions that increase the risk of hemorrhage (such as multiple pregnancies, coagulation defects, fibroids, and abnormal placental adhesions, such as placenta previa) were excluded from the study (Figure 1).

The cases (*n* = 216) were divided into two groups in sequential order according to the order of cesarean section:

Group 1 (*n* = 108): administration of oxytocin before removal of the placenta (AOBRP).

Group 2 (*n* = 108): administration of oxytocin after removal of the placenta (AOARP).

### 2.2. Intervention 

Cesarean sections were performed under spinal anesthesia by the same investigator in all cases in both groups. In all deliveries, the abdomen was entered with a Pfannenstiel incision, the fetus was removed from the uterus with a transverse incision of the lower segment of the uterus, and then the umbilical cord was clamped and cut immediately. The placenta was removed using the manual traction method in all cases. The standard dose of oxytocin is delivered as an intravenous (IV) push of 3 international units (IU); simultaneously, 10 IU of oxytocin is added to 1000 cc isotonic fluid and given as an IV infusion at a rate of 250 cc/h [24]. All methods and procedures applied to both groups were identical, except for the timing of administration of the standard dose of oxytocin. Adjustment and administration of the same dose of oxytocin in all cases were performed by anesthesiologists.

The timing of administration for the groups is given below.

AOBRP group: In all cases, the umbilical cord was clamped after the removal of the fetus. Immediately after clamping, 3 IU of oxytocin was given as an IV push before the placenta was removed. The placenta was removed by manual traction, while the fluid containing 10 IU of oxytocin in 1000 cc isotonic fluid was administered IV at a rate of 250 cc/h.AOARP group: In all cases, the umbilical cord was clamped after the fetus was removed. After clamping, the placenta was removed by manual traction. After removal of the placenta, 3 IU of oxytocin was given as an IV push. Simultaneously, 10 IU of oxytocin was added to 1000 cc of isotonic fluid and administered at a rate of 250 cc/h IV.

### 2.3. Measurements

As an additional uterotonic agent, an IV push of 5 IU of oxytocin and 0.2 mg methylergonovine maleate were administered intramuscularly (IM) to cases with excessive hemorrhage during cesarean section. The decision related to excessive hemorrhage during a cesarean was made by the responsible obstetrician who performed the surgery. A visible excessive bleeding, and a rapid increase in aspiration fluid were defined as excessive hemorrhage.

Two methods were used to measure the amount of intraoperative bleeding during cesarean section in all cases: (1) preoperative and immediate postoperative weights of compresses and sponges used to absorb intraoperative bleeding were weighed, and 1 g of the weight difference was converted to milliliters to be 1 mL; and (2) intraoperatively aspirated bleeding and aspiration fluid including amniotic fluid were measured and the estimated amniotic fluid amount (measured by ultrasonography just before cesarean section) was subtracted from this fluid. Thus, the sum of both methods was evaluated as intraoperative bleeding.

All cases were evaluated with obstetric USG before cesarean section. The fetal position, estimated fetal weight, placental localization, and estimated amount of amniotic fluid were measured. The height and weight of all cases were measured before cesarean section, and the body mass index (BMI) was calculated using these values (kg/m^2^). The hemoglobin (HB) and hematocrit (HTC) values were measured preoperatively, at the 6th hour and at the 24th hour postoperatively in all cases. The starting and ending times of the cesarean section were recorded for all cases, and the duration of the operation was calculated. All cases were followed up at the obstetrics clinic for approximately 48 h after cesarean section. During the first 48 h of hospitalization in the obstetrics clinic, 3 × 5 IU/day oxytocin was administered intramuscularly to all cases routinely, and the amount of bleeding was measured by counting the number of pads used.

### 2.4. Sample Size and Statistical Analysis 

According to the reference study results [25], they had a large effect size (d = 2.425). Assuming we can achieve an effect size at a small level (d = 0.4), a power analysis was performed before the study. Accordingly, when at least 200 participants (100 per group) were included in the study, that would result in 80% power with 95% confidence level (5% type 1 error rate). Statistical analysis was performed using SPSS version 25.0 software (IBM Corp., Armonk, NY, USA). Numerical variables are expressed as mean ± standard deviation and categorical variables as numbers and percentages (%). Numerical data were analyzed for normal distribution by skewness. An independent-sample *t*-test and a post hoc test were used to analyze the differences between the groups; *p* < 0.05 was set as the threshold for significance.

### 2.5. Ethical Approval

This study was conducted in accordance with the principles of the Declaration of Helsinki. All participants gave their written and informed consent prior to their participation in this study. Ethical approval for the study was obtained from the Pamukkale University Clinical Research Ethics Committee (14 December 2021; 22).

## 3. Results

Table 1 compares the descriptive parameters of the two groups. Age (year), BMI (kg/m^2^), parity, gestational week, surgical time, and newborn weight (g) did not differ between the groups (*p* = 0.406, 0.238, 0.704, 0.390, 0.399, and 0.141, respectively).

Table 2 compares the hemorrhage parameters of the two groups. Preoperative HB and HTC levels did not differ between the groups (*p* = 0.665 and 0.755, respectively). The estimated amount of intraoperative blood loss was higher in the AOARP group than in the AOBRP group (537.77 ± 113 mL vs. 425.83 ± 90 mL, respectively). Postoperative 6th hour HB and HTC and postoperative 24th hour HB and HTC levels were significantly higher in the AOBRP group than in the AOARP group (*p* = 0.002, 0.002, 0.002, and 0.001, respectively). Postoperative hemorrhage (3.33 ± 0.86 pads vs. 3.52 ± 0.89 pads, respectively), uterotonic requirement during cesarean section (5 (4.6%) cases vs. 12 (11.1%) cases, respectively), and the need for blood transfusion (1 (0.9%) case vs. 4 (3.7%) cases, respectively) did not differ between the AOBRP group and the AOARP group (*p* > 0.05).

Table 3 compares the hemoglobin levels at different times for each group. The mean HB difference between the preoperative and postoperative 6th hour was 0.524 g/dL in the AOBRP group, whereas it was 1.061 g/dL in the AOARP group (*p* = 0.026 and 0.000, respectively). Similarly, the mean hemoglobin difference between the preoperative and postoperative 24th hour was lower in the AOBRP group than in the AOARP group (0.626 g/dL vs. 1.153 g/dL; *p* = 0.004 and 0.000, respectively). The mean hemoglobin difference between the postoperative 6th hour and the postoperative 24th hour was 0.102 g/dL in the AOBRP group and 0.092 g/dL in the AOARP group; these differences were not statistically significant (*p* = 0.936 and 0.954, respectively).

## 4. Discussion 

In the current study, we aimed to determine the effects of the timing of standard-dose oxytocin administration (which we routinely apply during cesarean sections in our clinic) on the amount of intrapartum and postpartum bleeding. Therefore, we administered oxytocin at two different times: before and after placenta removal. The main finding was that there was less intraoperative bleeding and less decrease in postoperative HB and HTC levels with oxytocin administration before placenta removal.

Oxytocin is still the drug of choice for the prevention and treatment of PPH. According to WHO recommendations, oxytocin (10 IU, IM, or IV) is the recommended uterotonic agent for the prevention of PPH in all deliveries in all settings [16]. If PPH occurs, additional units (up to a total of 40 IU) should be administered intravenously until the hemorrhage stops [16].

There are many studies in the literature on the dose of oxytocin administration [16,24,26,27,28,29]. However, studies on the timing of oxytocin administration are limited [30]. In this study, the effects of the timing of oxytocin administration on PPH were investigated. We found that the descriptive features of the research groups did not differ statistically. This result shows that the groups are homogeneous and that the results of the comparisons are consistent. In addition, since the preoperative hemoglobin and hematocrit levels did not differ between the groups, the validity and reliability of the postoperative results are enhanced.

Ahmadi [31] stated that the use of 80 units of oxytocin in the prevention of uterine atony after cesarean section resulted in a decrease in uterine atony and a decrease in the need for an additional uterotonic drug compared with a dose of 30 units of oxytocin. However, it was stated that the dose did not have a significant effect on the rate of decrease in hemoglobin at 6 and 24 h after surgery. In the current study, we administered an equal dose of oxytocin (an IV push of 3 IU and an IV infusion of 250 cc/h with a fluid containing 10 IU of oxytocin in 1000 cc) in both groups to compare the effects of oxytocin with respect to the timing of administration.

Mangla et al. [32] administered oxytocin (5 IU/10 mL of saline) directly to the myometrium after the fetus was born (*n* = 50) or without placental separation (*n* = 50). According to their results, intramyometrial oxytocin injection before placenta separation was effective in increasing uterine contraction and reducing the incidence of PPH. In our study, we administered oxytocin intravenously (only).

Takmaz et al. [33] evaluated the positive effect of IV oxytocin infusion on intraoperative blood loss in the early period before uterine incision. They concluded that early initiation of IV oxytocin infusion from the uterine incision is more effective in reducing intraoperative blood loss than the late infusion of IV oxytocin after umbilical cord clamping or delivery of the placenta. If oxytocin is started before the delivery of the baby, fetal well-being should be well monitored during the delivery, whereas Takmaz et al. recorded only the postnatal APGAR scores of the babies in their study and reported that there was no difference between the two groups in terms of APGAR scores. In our study, we administered an equal dose of oxytocin two separate times immediately after the birth of the baby: before and after the placenta was removed. Postoperative 6th-hour HB and HTC and postoperative 24th hour HB and HTC values decreased more in the OARP group than in the OBRP group, according to the preoperative values. However, postoperative 24th hour hemorrhage (number of pads), uterotonic requirement during CS, and the need for blood transfusion did not change according to the time of oxytocin administration.

Cecilia et al. [29] compared one such protocol using single-dose IV oxytocin over 2–4 h (total = 10 units) with oxytocin maintenance infusion for 8–12 h (total = 30 units) in postoperative CS women to prevent PPH. When their results were analyzed, it was found that both regimens were equally effective in the prevention of PPH in postoperative CS women. Both treatment regimens were associated with a similar amount of blood loss during the operative and postoperative periods. Thus, they reported that the low-dose oxytocin regimen was as effective as a high-dose oxytocin regimen in the prevention of PPH in postoperative CS women. In our current study, we only changed the timing of oxytocin administration without making any changes in oxytocin doses. In our current study, we only changed the timing of oxytocin administration without making any changes in oxytocin doses. Thus, we found that oxytocin administered before removal of the placenta after clamping the umbilical cord was significantly effective on intraoperative and postoperative bleeding in CS.

Torloni et al. [30] stated that the earlier prophylactic administration of oxytocin in CS may be slightly more beneficial than subsequent administration (i.e., after fetal delivery) without an increase in side effects. In addition, oxytocin given before fetal delivery significantly reduced intraoperative blood loss, but did not change the incidence of blood transfusion. Similarly, in our study, intraoperative blood loss was lower in the group that received oxytocin before placenta removal.

In a study by Tharwat et al. [34], 300 elective cesarean section patients were divided into two groups, and oxytocin was administered during anesthesia induction and after delivery. The study stated that oxytocin given as an IV infusion during anesthesia induction before skin incision during CS is more effective in reducing blood loss and preventing PPH compared to oxytocin administration after delivery of the fetus. However, in both studies, in which oxytocin administration was initiated before the birth of the baby, there were no detailed data on how fetal well-being was monitored during delivery or whether postpartum umbilical cord blood gases and APGAR scores of babies were recorded.

There is a need for additional, well conducted, and well reported, trials on the timing of prophylactic oxytocin in women giving birth by CS, to increase the overall certainty of the evidence on this important clinical question. Ideally, future RCTs should be placebo-controlled and double-blinded, involve other obstetric populations (women with previous CS and those at high risk for PPH), as well as other types of CS (in the 1st and 2nd stages of spontaneous and induced labor previously exposed to oxytocin), and measure all PPH prevention core results, including adverse effects and women’s views.

The following study limitations were identified: (1) it was a single-center study; (2) a group of spontaneous vaginal deliveries was not formed; (3) a group with emergency cesarean sections other than elective cesarean sections was not formed; and (4) only cesarean sections at 38–40 weeks of gestation were evaluated.

## 5. Conclusions

In conclusion, avoiding PPH is vital for both maternal and newborn health. The timing of standard-dose prophylactic oxytocin administration significantly altered the amount of intraoperative bleeding and the postoperative 6th and 24th hour HB and HTC values in the current study. Administration of oxytocin before removal of the placenta resulted in less intraoperative bleeding and less decrease in postoperative HB and HTC values. We believe that the main reason for these changes may be that administration of oxytocin before placenta removal leads to uterine contraction with an earlier effect following placental removal. The main strength of our study was a uniform treatment protocol as the study was conducted in a single center and appropriate sample size. Further studies are needed to elucidate this situation.

## Figures and Tables

**Figure 1 medicina-59-00222-f001:**
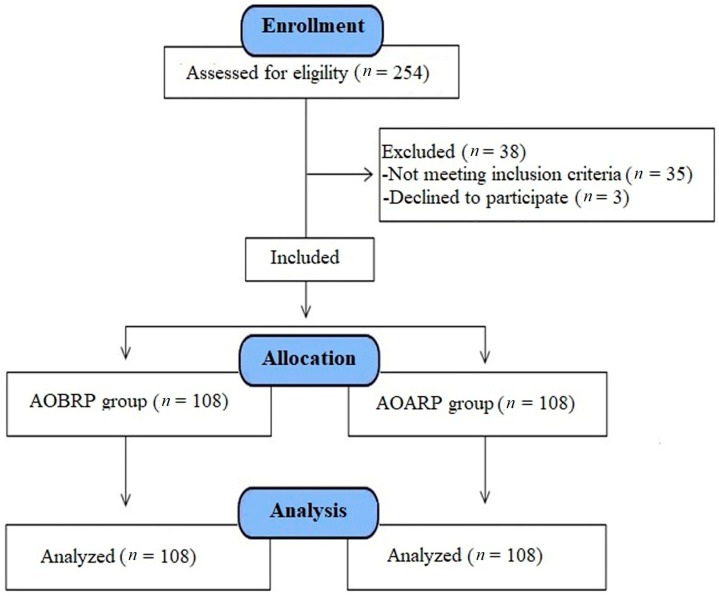
Flow consort chart of the study.

**Table 1 medicina-59-00222-t001:** Comparison of the descriptive parameters between the two groups.

Parameter	AOBRP (*n* = 108)	AOARP (*n* =108)	*p **
	Mean ± SD	Mean ± SD	
Age (year)	30.66 ± 3.93	31.13 ± 4.39	0.406
BMI (kg/m^2^)	30.48 ± 2.48	30.91 ± 2.81	0.238
Parity	2.02 ± 0.75	1.99 ± 0.67	0.704
Gestational week	38.62 ± 0.74	38.71 ± 0.83	0.390
Surgical time (min)	38.08 ± 10.05	39.21 ± 9.56	0.399
Newborn weight (g)	3278.24 ± 261.49	3225 ± 268.56	0.141

** p* < 0.05 is considered statistically significant; AOBRP: administration of oxytocin before removal of the placenta; AOARP: administration of oxytocin after removal of the placenta; BMI: body mass index; min: minute; g: gram.

**Table 2 medicina-59-00222-t002:** Comparison of the hemorrhage parameters between the two groups.

Parameter		AOBRP (*n* = 108)	AOARP (*n* = 108)	*p* *
		Mean ± SD	Mean ± SD	
Preop. HB (g/dL)		12.16 ± 1.43	12.08 ± 1.28	0.665
Preop. HTC (%)		35.92 ± 4.33	35.75 ± 3.79	0.755
Postop. 6th hour HB (g/dL)		11.64 ± 1.46	11.02 ± 1.45	0.002 *
Postop. 6th hour HTC (%)		34.50 ± 4.35	32.60 ± 4.37	0.002 *
Postop. 24th hour HB (g/dL)		11.54 ± 1.41	10.93 ± 1.45	0.002 *
Postop. 24th hour HTC (%)		34.20 ± 4.19	32.33 ± 4.30	0.001 *
Intraoperative hemorrhage (mL)		425.83 ± 90.24	537.77 ± 113.17	0.000 *
Postop. hemorrhage (number of pads)		3.33 ± 0.86	3.52 ± 0.89	0.114
Uterotonic requirement during caesarean section	No	103 (95.4%)	96 (88.9%)	0.077 (χ² = 3.129)
	Yes	5 (4.6%)	12 (11.1%)	
Need for blood transfusion	No	107 (99.1%)	104 (96.3%)	0.175 (χ² = 1.843)
	Yes	1 (0.9%)	4 (3.7%)	

* *p* < 0.05 is considered statistically significant; Preop.: preoperative; Postop.: postoperative; HB: hemoglobin; HTC: hematocrit; AOBRP: administration of oxytocin before removal of the placenta; AOARP: administration of oxytocin after removal of the placenta.

**Table 3 medicina-59-00222-t003:** Comparison of the hemoglobin levels at different times for each group.

			Mean Difference	Std. Error	*p* *
AOBRP (*n* = 108)	Preop. HB (g/dL)	Postop. 6th hour HB (g/dL)	0.524	0.19	0.026 *
		Postop. 24th hour HB (g/dL)	0.626	0.9	0.004 *
	Postop. 6th hour HB (g/dL)	Postop. 24th hour HB (g/dL)	0.102	0.19	0.936
AOARP (*n* = 108)	Preop. HB (g/dL)	Postop. 6th hour HB (g/dL)	1.061	0.18	0.000 *
		Postop. 24th hour HB (g/dL)	1.153	0.18	0.000 *
	Postop. 6th hour HB (g/dL)	Postop. 24th hour HB (g/dL)	0.092	0.19	0.954

* *p* < 0.05 is considered statistically significant; Preop.: preoperative; Postop.: postoperative; HB: hemoglobin; HTC: hematocrit; AOBRP: administration of oxytocin before removal of the placenta; AOARP: administration of oxytocin after removal of the placenta.

## Data Availability

The datasets used and/or analyzed during the current study are available from the corresponding author on reasonable request.

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
