# Peer review of "Investigation of The Effects of Oxytocin Administration Timing on Postpartum Hemorrhage during Cesarean Section"

_medicina, 2023, doi:10.3390/medicina59020222_

Round 1

Reviewer 1 Report

I suggest the authors to give some data reading C-Section rate globally (example: Betrán AP, Ye J, Moller AB, et al. The Increasing Trend in Caesarean Section Rates: Global, Regional and National Estimates: 1990-2014. PLoS One. 2016; 11(2): e0148343. Or https://www.who.int/news/item/16-06-2021-caesarean-section-rates-continue-to-rise-amid-growing-inequalities-in-access).

You have applied manual traction method. Why did you choose manual traction method instead of manual removal method? Or what were the reasons for avoiding combined approach (manual traction method in some patients and manual removal method in other patients)?

Some parts of the Methods section are not so precise. For example, you have applied subjective nature of visual estimation of blood loss:

- You have written "A sudden increase in the number of compresses". Please define what sudden increase is. Sudden – define, increase – define

- "visible excessive bleeding

- "rapid increase in aspiration fluid"

Furthermore, this subjective nature of visual estimation of blood loss is even potentiated with the fact that only one investigator have performed all 216 C-Sections (You have written “Cesarean sections were performed under spinal anesthesia by THE SAME investigator in all cases in both groups" and “The decision related to excessive hemorrhage during a cesarean was made BY THE RESPONSIBLE OBSTETRICIAN WHO PERFORMED SURGERY". That raises the issues of inter-observer and intra-observer variability in interpretation of your results, of future application for clinical practice and of future research. What is power of study? Have you performed sample size calculation?

The sentences “PPH is the leading cause of maternal death worldwide [21]. The WHO estimates that approximately 303,000 maternal deaths occur each year, of which approximately 20% are due to PPH. PPH is both preventable and treatable, with widely available and inexpensive drugs." belong to Introduction section, not for Discussion. I suggest the authors to begin by briefly answering all of the research questions underpinning the problem that you posed in the introduction.

Author Response

Dear Reviewer, thank you for your interest in our article. We thank to for your contubition to improve our paper. We have revised our manuscript according to the comments of the Reviewer 1 and Reviewer 2, and the revisions were made in red style. We had Scribendi do the editing and proofreading of this manuscript. We have prepared the answers given to comments of reviewers in below.    

Reviewer: 1

Comments to the Author

You have applied manual traction method. Why did you choose manual traction method instead of manual removal method? Or what were the reasons for avoiding combined approach (manual traction method in some patients and manual removal method in other patients)?

Response to Reviewer: 1

We used the same method for removal of the placenta in all cases included in our study so that the amount of bleeding due to the removal methods of the placenta is not affected.

Reviewer: 1

Some parts of the Methods section are not so precise. For example, you have applied subjective nature of visual estimation of blood loss:

- You have written "A sudden increase in the number of compresses". Please define what sudden increase is. Sudden – define, increase – define

- "visible excessive bleeding”

- "rapid increase in aspiration fluid"

Response to Reviewer: 1

I meant that the frequency of replacement of the compresses used to absorb intraoperative bleeding is increased in cases where the amount of bleeding is high.

Reviewer: 1

Furthermore, this subjective nature of visual estimation of blood loss is even potentiated with the fact that only one investigator have performed all 216 C-Sections (You have written “Cesarean sections were performed under spinal anesthesia by THE SAME investigator in all cases in both groups" and “The decision related to excessive hemorrhage during a cesarean was made BY THE RESPONSIBLE OBSTETRICIAN WHO PERFORMED SURGERY". That raises the issues of inter-observer and intra-observer variability in interpretation of your results, of future application for clinical practice and of future research. What is power of study? Have you performed sample size calculation?

Response to Reviewer: 1

We did a power analysis of our study. I placed the power analysis and sample study in the Sample size and statistical analysis section of the material and method section of the manuscript.

Reviewer: 1

The sentences “PPH is the leading cause of maternal death worldwide [21]. The WHO estimates that approximately 303,000 maternal deaths occur each year, of which approximately 20% are due to PPH. PPH is both preventable and treatable, with widely available and inexpensive drugs." belong to Introduction section, not for Discussion. I suggest the authors to begin by briefly answering all of the research questions underpinning the problem that you posed in the introduction.

Response to Reviewer: 1

This section has been removed from the discussion section.

Reviewer 2 Report

This article evaluated the effects of oxytocin administration timing on postpartum hemorrhage during cesarean section. There are some issues in this manuscript that should be addressed as follows:

1.    Title: The word “Sesarean” should be replaced with “cesarean”.

2.    Abstract: The heading “Background and Objectives” should be replaced with “Objectives”.

3.    Abstract: The headingMaterials and Methodsshould be replaced with “Patients and Methods”.

4.    The novel points in this article should be clarified because there are previous reports that discussed similar issues; e.g. https://pubmed.ncbi.nlm.nih.gov/34081734/; https://pubmed.ncbi.nlm.nih.gov/30636832/

5.    Page 2 Lines 66-67: The sentence “PPH, one of the most common causes of maternal morbidity and mortality, increases in parallel with an increase in cesarean section rates.” has no reference. Please, add a reference.

6.    The method of calculation of the appropriate sample size of the participants should be mentioned.

7.    Page 3: References for the utilized doses of oxytocin should be added.

8.    Page 3: A reference for the methods used to measure the amount of intraoperative bleeding during cesarean section should be added.

9.    Page 3: References for the methods of measurement of hemoglobin (HB) and hematocrit (HTC) values should be added.

10. A diagram summarizing the main findings of the study should be added.

11. Page 6 Lines 208 and 209: The sentence “There are many studies in the literature on the dose of oxytocin administration. However, studies on the timing of oxytocin administration are limited.” has no references. Please, add a reference.

12. Discussion: The discussion should discuss in a more detailed manner the main findings of the present study.

13. More recent references should be added to the discussion because the number of 26 references is too small for a research article.

14. I think that the conclusion is not sufficient. It should delineate the possible clinical implications of the data obtained from the present study.

15. The manuscript should be thoroughly checked regarding the grammatical and typing errors.

Author Response

Dear Reviewer, thank you for your interest in our article. We thank to the reviewers for their contubition to improve our paper. We have revised our manuscript according to the comments of the Reviewer 1 and Reviewer 2, and the revisions were made in red style. We had Scribendi do the editing and proofreading of this manuscript. We have prepared the answers given to comments of reviewers in below.    

Reviewer: 2

Comments to the Author

  1. Title: The word “Sesarean” should be replaced with “cesarean”.
  2. Abstract: The heading “Background and Objectives” should be replaced with “Objectives”.
  3. Abstract: The heading “Materials and Methods” should be replaced with “Patients and Methods”.
  4. The novel points in this article should be clarified because there are previous reports that discussed similar issues; e.g. https://pubmed.ncbi.nlm.nih.gov/34081734/; https://pubmed.ncbi.nlm.nih.gov/30636832/
  5. Page 2 Lines 66-67: The sentence “PPH, one of the most common causes of maternal morbidity and mortality, increases in parallel with an increase in cesarean section rates.” has no reference. Please, add a reference.
  6. The method of calculation of the appropriate sample size of the participants should be mentioned.
  7. Page 3: References for the utilized doses of oxytocin should be added.
  8. Page 3: A reference for the methods used to measure the amount of intraoperative bleeding during cesarean section should be added.
  9. Page 3: References for the methods of measurement of hemoglobin (HB) and hematocrit (HTC) values should be added.
  10. A diagram summarizing the main findings of the study should be added.
  11. Page 6 Lines 208 and 209: The sentence “There are many studies in the literature on the dose of oxytocin administration. However, studies on the timing of oxytocin administration are limited.” has no references. Please, add a reference.
  12. Discussion: The discussion should discuss in a more detailed manner the main findings of the present study.
  13. More recent references should be added to the discussion because the number of 26 references is too small for a research article.
  14. I think that the conclusion is not sufficient. It should delineate the possible clinical implications of the data obtained from the present study.
  15. The manuscript should be thoroughly checked regarding the grammatical and typing errors.

Response to Reviewer: 2

  1. The word “sesarean” was replaced to “cesarean”.
  2. Abstract: The heading of "Background and Objectives" has been changed to "Objectives".
  3. Abstract: The heading “Materials and Methods” has been changed to “Patients and Methods”.
  4. Bu makaledeki yeni noktalar açıklığa kavuşturulmaya çalışıldı.
  5. References have been added.
  6. Appropriate sample size and power analysis of the participants were done beforehand. I placed the power analysis and sample study in the Sample size and statistical analysis section of the material and method section of the manuscript.
  7. References have been added.
  8. Reference added.
  9. References have been added.
  10. The flow consort chart of the study has been added.
  11. References have been added.
  12. The discussion has been rearranged.
  13. More references have been added to the discussion.
  14. New additions have been made to the Conclusion section.
  15. The manuscript has been rechecked for grammatical and spelling errors.

Round 2

Reviewer 1 Report

Dear authors you have successfully answered all my questions and queries.

Author Response

Dear Reviewer, thank you for your interest in our article. We thank to for your contubition to improve our paper.

Reviewer: 1

Dear authors you have successfully answered all my questions and queries.

Response to Reviewer: 1

Thank you for your comment on our study.

Reviewer 2 Report

The authors had appropriately addressed most of my comments. However, they didn't clarify the novel points in this article because there are previous reports that discussed similar issues; e.g. https://pubmed.ncbi.nlm.nih.gov/34081734/; https://pubmed.ncbi.nlm.nih.gov/30636832/. The novel points should be clarified in the "Introduction" section.

Author Response

Dear Reviewer, thank you for your interest in our article. We thank to the reviewers for their contubition to improve our paper. We have revised our manuscript according to the comments of the Reviewer 2, and the revisions were made in red style.  

Reviewer: 2

Comments to the Author

The authors had appropriately addressed most of my comments. However, they didn't clarify the novel points in this article because there are previous reports that discussed similar issues; e.g. https://pubmed.ncbi.nlm.nih.gov/34081734/; https://pubmed.ncbi.nlm.nih.gov/30636832/. The novel points should be clarified in the "Introduction" section.

Response to Reviewer: 2

New points in this article have been tried to be clarified in the introduction.
